# Human leukocyte antigen-*DQA1\*04:01* and rs2040406 variants are associated with elevated risk of childhood Burkitt lymphoma

Zhiwei Liu [1,29], Yang Luo [2,3,4,5,6,7,29], Samuel Kirimunda[8], Murielle Verboom[9], Olusegun O. Onabajo [1], Mateus H. Gouveia [10], Martin D. Ogwang[11,12], Patrick Kerchan[12,13], Steven J. Reynolds[14], Constance N. Tenge[12,15], Pamela A. Were[12,16], Robert T. Kuremu[12,15], Walter N. Wekesa[12,15], Nestory Masalu[17], Esther Kawira[12,18], Tobias Kinyera[11,12], Isaac Otim[11,12], Ismail D. Legason [12,13], Hadijah Nabalende[11,12], Herry Dhudha[12,18], Leona W. Ayers[19], Kishor Bhatia[1], James J. Goedert[1], Nathan Cole[20], Wen Luo[20], Jia Liu[20], Michelle Manning[20], Belynda Hicks [20], Ludmila Prokunina-Olsson [1], George Chagaluka[21], W. Thomas Johnston[22], Nora Mutalima[22,23], Eric Borgstein[21], George N. Liomba[21], Steve Kamiza[21], Nyengo Mkandawire[21], Collins Mitambo[24], Elizabeth M. Molyneux[21], Robert Newton [22], Ann W. Hsing[25], James E. Mensah[26], Anthony A. Adjei[26], Amy Hutchinson[20], Mary Carrington [27,28], Meredith Yeager [20], Rainer Blasczyk [9], Stephen J. Chanock [1], Soumya Raychaudhuri [3,4,5,6,7,30] & Sam M. Mbulaiteye [1,30✉]

Burkitt lymphoma (BL) is responsible for many childhood cancers in sub-Saharan Africa, where it is linked to recurrent or chronic infection by Epstein-Barr virus or *Plasmodium falciparum*. However, whether human leukocyte antigen (*HLA*) polymorphisms, which regulate immune response, are associated with BL has not been well investigated, which limits our understanding of BL etiology. Here we investigate this association among 4,645 children aged 0-15 years, 800 with BL, enrolled in Uganda, Tanzania, Kenya, and Malawi. *HLA* alleles are imputed with accuracy >90% for *HLA* class I and 85-89% for class II alleles. BL risk is elevated with *HLA-DQA1\*04:01* (adjusted odds ratio [OR] = 1.61, 95% confidence interval [CI] = 1.32-1.97, $P = 3.71 \times 10^{-6}$), with rs2040406(G) in *HLA-DQA1* region (OR = 1.43, 95% CI = 1.26-1.63, $P = 4.62 \times 10^{-8}$), and with amino acid Gln at position 53 versus other variants in *HLA-DQA1* (OR = 1.36, $P = 2.06 \times 10^{-6}$). The associations with *HLA-DQA1\*04:01* (OR = 1.29, $P = 0.03$) and rs2040406(G) (OR = 1.68, $P = 0.019$) persist in mutually adjusted models. The higher risk rs2040406(G) variant for BL is associated with decreased *HLA-DQB1* expression in eQTLs in EBV transformed lymphocytes. Our results support the role of *HLA* variation in the etiology of BL and suggest that a promising area of research might be understanding the link between *HLA* variation and EBV control.

---

A full list of author affiliations appears at the end of the paper.

Burkitt lymphoma (BL) is an aggressive B cell non-Hodgkin lymphoma[1] first reported in African children in 1958 by Denis Burkitt[2]. BL incidence varies 3- to 5-fold within and across continents, but the highest incidence is recorded in children in sub-Saharan Africa (SSA, 2-4 per 100,000 person-years[3,4]), where it is approximately 10-fold higher than the incidence in the United States or Europe[1]. Epstein-Barr virus (EBV)[5] and *Plasmodium* (*P.*) *falciparum*[6] are established risk factors for BL in SSA. The association of BL with these infections strongly suggests that variation within the human leukocyte antigen (HLA) region, which plays an important role in the regulation of immune response[7], could be related to the etiology of BL[8]. This hypothesis is consistent with studies that showed association of *HLA-B*53* with protection against severe malaria in Gambian children[9]. Mechanistic studies suggested that this association was mediated by enhanced cytotoxicity against liver-stage malaria parasite forms among carriers of *HLA-B*53*[10]. However, a large genome-wide association study (GWAS) of 17,000 individuals from 11 populations, including people from Gambia, did not replicate this association[11]. HLA variation has also been associated with EBV control, including *HLA-A*02:01*[12] and *HLA-DQB1*02*[13] which are associated with elevated anti-EBV antibodies. *HLA-A*02:01* is part of the HLA Class I system and may mediate direct killing of EBV-infected cells by modulating expression of CD8+ Th1-type immune response[14]. Conversely, *HLA-DQB1*02* is part of the HLA Class II system and may mediate EBV control by facilitating anti-EBV antibody secretion through the modulation of peptide presentation to CD4+ T cells and their Th-2 signaling to B cells, thereby promoting an antibody response[15,16]. Therefore, the reported HLA associations are compatible with HLA modulating immune response to EBV or malaria as potential mechanisms for influencing risk of BL.

Only five studies have been conducted to directly test the association between HLA variation and risk of BL, yielding null results[8,17–20]. Those studies were small (<100 cases), lacked suitable controls, and did not properly adjust for ancestry of participants. Moreover, most typed HLA alleles using serological methods, which can be less accurate[21] and are limited to 1-field resolution. Only one study, which was conducted by our group previously, has used sequence-based typing (SBT)[22] to obtain accurate high-resolution data (≥2 fields) in 600 participants (including 200 with BL) in the Epidemiology of Burkitt Lymphoma in East African Children and Minors (EMBLEM) study in Uganda[23]. Potentially significant associations were found between BL and *HLA-A*02*, *HLA-B*41*, and *HLA-B*58*[23], underscoring the need to conduct HLA research using high-resolution typing and in a larger sample. We investigated associations of classical HLA alleles, SNPs, and amino acids within the HLA region with BL among 4645 children, 800 with BL, enrolled in Uganda, Tanzania, Kenya, and Malawi in who HLA variation was inferred from genome-wide genotype data obtained from an ongoing BL GWAS[24]. Here we report successful HLA imputation, using an updated multi-ancestry reference panel[25], obtaining accurate HLA imputation (>90% for class I and 85–89% for class II alleles) when compared with SBT-results in a subset of participants (n = 600) with paired imputed and SBT data[25]. We observe significant associations between BL risk and *HLA-DQA1*04:01* and rs2040406(G) in SSA. The higher risk rs2040406(G) variant for BL is associated with decreased *HLA-DQB1* expression in eQTLs in EBV transformed lymphocytes, potentially suggesting that the HLA role is mediated by EBV control.

## Results

Table 1 shows the participants' characteristics (see Methods and Fig. 1). Most participants were from the EMBLEM study in Uganda, Tanzania, and Kenya (71.9% of the cases and 94.5% of controls). As expected and by study design[1], compared to controls,

BL cases were predominantly male (62.9% vs. 52.1%) and were aged 3–11 years old (77.9% of the cases vs. 74.5% of the controls). *P. falciparum* infection was detected in 282 (35.3%) of the BL cases vs. 1857 (48.3%) of the controls[24].

HLA imputation accuracy at 2-field resolution for alleles with allelic fraction (AF) > 1% was >90% for Class I *HLA-A*, *-B*, and Class II *HLA-DQA1*, 89.5% for *HLA-DQB1*, 88.3% for *HLA-DPB1*, and 85.0% for *HLA-DRB1* (Supplementary Fig. 1A). We observed a high correlation ($r^2 \geq 0.8$) between HLA allelic dosage (genotype imputation quality info) and actual typed HLA alleles with AF ≥ 1% (Supplementary Fig. 1B). Analysis of ancestry using only GWAS SNPs in the HLA region replicated the ancestry patterns observed using genome-wide data[26] (Supplementary Fig. 2).

The number of imputed HLA variants after quality control is shown in Supplementary Table 1.

Of 12 HLA alleles with prior associations with severe malaria, EBV, or BL (Supplementary Table 2), nine with AF ≥ 1% were evaluated. *HLA-A*23:01*, which previously was associated with elevated anti-EBV VCA-IgG in a GWAS in Uganda[27], was associated with decreased BL in our study (OR = 0.77; P = 0.027; Supplementary Table 3). The other alleles, including *HLA-B*53*, were not associated with BL. In Uganda only, decreased BL risk

---

**Table 1 Demographic characteristics of 4645 participants in EMBLEM and Malawi.**

| Characteristics | Cases, n (%) (N = 800) | Controls, n (%) (N = 3845) |
|---|---|---|
| Country where enrolled | | |
| EMBLEM study | | |
| Uganda | 249 (31.1%) | 2004 (52.1%) |
| Tanzania | 97 (12.1%) | 753 (19.6%) |
| Kenya | 229 (28.6%) | 878 (22.8%) |
| Sub-total | 575 (71.9%) | 3,635 (94.5%) |
| Malawi | 225 (28.1%) | 210 (5.5%) |
| Sex[a] | | |
| Female | 297 (37.1%) | 1840 (47.9%) |
| Male | 503 (62.9%) | 2005 (52.1%) |
| Age, years | | |
| 0–2 | 67 (8.4%) | 342 (8.9%) |
| 3–5 | 219 (27.4%) | 914 (23.8%) |
| 6–8 | 239 (29.9%) | 1130 (29.4%) |
| 9–11 | 165 (20.6%) | 820 (21.3%) |
| 12–15 | 110 (13.8%) | 639 (16.6%) |
| *Plasmodium falciparum* positivity[b] | | |
| Negative | 499 (62.4%) | 1943 (50.5%) |
| Positive | 282 (35.3%) | 1857 (48.3%) |
| Missing | 19 (2.4%) | 45 (1.2%) |
| Season[c] | | |
| Dry | 115 (14.4%) | 1766 (45.9%) |
| Wet | 383 (47.9%) | 1869 (48.6%) |
| Missing/unknown | 302 (37.8%) | 210 (5.5%) |
| Area[d] | | |
| Urban | 216 (27.0%) | 1346 (35.0%) |
| Rural | 283 (35.4%) | 2289 (59.5%) |
| Missing/unknown | 301 (37.6%) | 210 (5.5%) |

[a]If the sex reported on the clinical forms did not match the genetically determined sex, the participant's sex was determined based on their genetic sex.
[b]*Plasmodium falciparum* positivity was based on parasitemia measured on thick blood film microscopy, antigenemia on rapid diagnostic tests, or on polymerase chain reaction tests (see methods).
[c]Data for season available only in the EMBLEM study. Calendar months of April–June and September–December were wet season months; calendar months of January–March and July–August were dry season months, based on reports produced by the National Bureau of Statistics of each country.
[d]Rural/urban status was determined only for the EMBLEM study. Rural areas were defined as "low-population density" villages if the population count of children aged 0–15 years in the village was less than the average population count for census enumeration areas in the regions studied for each country; otherwise as "urban/high-population density".

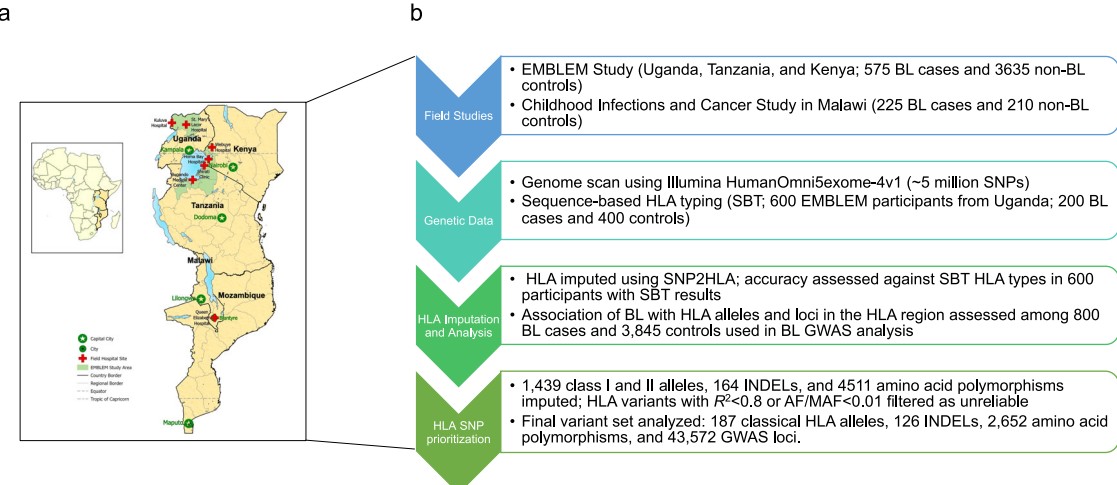

**Fig. 1 Map of study area and flow chard of study procedures. a** Map showing geographical areas shaded green where Burkitt lymphoma cases and controls were enrolled. Participating hospitals are marked with a red cross, while the capital cities where major tertiary care centers are located are marked with white star on a green background. The map was drawn using ESRI ArcGIS Pro software. No portions of this figure were imported as image components from a database. **b** The Study workflow components.

was observed with *HLA-A*02* (OR = 0.61; *P* = 0.001) and *-B*41* (OR = 0.45; *P* = 0.037; Supplementary Table 4), which in line with the findings we previously reported in a smaller sample size in Uganda.

After Bonferroni correction (*P* < 2.7 × 10$^{-4}$), *HLA-DQA1*04:01* was associated with elevated risk of BL (OR = 1.61, 95% CI = 1.32–1.97; *P* = 3.71 × 10$^{-6}$, Table 2, Fig. 2A). Significant associations were observed with *DQA1*04* (OR = 1.60, 95% CI = 1.31–1.96; *P* = 5.02 × 10$^{-6}$), *DQB1*04* (OR = 1.60, 95% CI = 1.30–1.98; *P* = 1.29 × 10$^{-5}$), and *DRB1*03:02* (OR = 1.47, 95% CI = 1.20–1.81; *P* = 2.30 × 10$^{-4}$) (Supplementary Table 5). These associations were not significant in analysis conditioned on *HLA-DQA1*04:01* (Fig. 2B). The observed associations were similar in country-specific analyses (Supplementary Table 6) and in the different sensitivity models (Supplementary Fig. 3 for model 1 and Supplementary Table 7 for models 2-5).

We identified 24 haplotypes formed by *HLA-DRB1, -DQA1* and *-DQB1* genes with AF ≥ 0.01. The *HLA* haplotype substructure was different in our participants in East Africa compared to those in Ghana, consistent with our findings of significant population substructure between these regions[26] (Supplementary Fig. 4). For example, *HLA* haplotype *DRB1*08:04-DQB1*03:01-DQA1*04:01* was more frequent in Ghana (5%; 45/923) compared to East Africa (1%; 32/4645). This haplotype was not associated with BL risk (OR = 1.44, 95% CI = 0.82–2.51; *p* = 0.200), whereas *DRB1*03:02-DQB1*04:02-DQA1*04:01*, which was also more frequent in Ghana (10.0%), was associated with elevated BL risk (OR = 1.58, 95% CI = 1.25–1.99; *P* = 1.08 × 10$^{-4}$) (Table 2).

After Bonferroni correction (*P* < 1.1 × 10$^{-6}$), GWAS variant rs2040406(G) in the *HLA-DQA1* region was associated with elevated BL risk (OR = 1.43, 95% CI = 1.26–1.63; *P* = 4.62 × 10$^{-8}$, Table 2, Fig. 2A, and Supplementary Fig. 5). This association persisted in single variant conditional models adjusting for *HLA-DQA1*04:01* (Table 2). Associations with elevated BL risk were observed with rs1064994(C) (OR = 1.38, *P* = 8.75 × 10$^{-7}$), rs1065049(A) (OR = 1.39, *P* = 6.06 × 10$^{-7}$), rs9272982(A) (OR = 1.38, *P* = 8.23 × 10$^{-7}$), and rs1130399(A) (OR = 1.48, *P* = 1.21 × 10$^{-7}$), but they were not significant after conditioning on rs2040406, which may reflect LD patterns that were strong (*r*$^2$ > 0.8) for three of the four SNPs and weak or moderate for rs1130399 (*r*$^2$ = 0.3) (Fig. 2C and Supplementary Table 8). Finally, we examined the combined effect of the presence of the two risk alleles. Although homozygosity for both

*HLA-DQA1*04:01* and SNP rs2040406 (with LD R$^2$ = 0.225) was low (1.3% in the cases and 0.68% in the controls), children who were homozygous for both had a higher risk for BL (OR = 2.66, 95% CI = 1.19–5.95; *P* = 0.017) versus those not carrying either allele.

Among the imputed single amino acid residues in HLA-DQA1 chain, significant associations were observed with variants at position 53, with two allelic variants (Lys/Gln) in the current dataset. Compared to Lys, which is the most common residue at position 53, those having Gln residue at this position had an elevated risk for BL (OR = 1.36; 95% CI = 1.20–1.55; *P* = 2.06 × 10$^{-6}$; Table 2). The 3D structure of the HLA-DQA1 chain suggests that Gln 53 is located within the peptide binding groove of HLA-DQ molecule, and may have functional impact on specificity of peptide binding and/or T cell receptor (TCR) contacts (Supplementary Fig. 6).

Among the controls, neither *HLA-DQA1*04:01* nor rs2040406(G) alleles were associated with *P. falciparum* parasite density (Supplementary Fig. 7). In exploratory analyses, we observed no significant associations between *HLA* alleles and *P. falciparum* infection detection among the controls (Supplementary Fig. 8).

Regarding a potential EBV connection with *HLA* in our data, we note suggestive associations of elevated BL risk (below Bonferroni threshold) were observed with three GWAS SNPs previously associated with higher anti-EBV EBNA1 IgG antibodies in southwest Uganda[27] (rs1064991(G), *P* = 2.11 × 10$^{-6}$; rs3129867(G), *P* = 8.24 × 10$^{-3}$; rs6927022(G), *P* = 1.99 × 10$^{-2}$; Supplementary Table 9). We found no association between BL with three other GWAS SNPs (all *p* values > 0.05) that were previously associated with higher anti-EBV EBNA1 IgG antibodies in Europeans (rs2516049)[28] and Mexican Americans (rs477515 and rs2854275)[29] or with GWAS SNP rs28394498(T) associated with anti-EBV VCA IgG antibodies in southwest Uganda[27] (*P* = 0.256). We note that the GWAS SNP with the strongest association with BL - rs2040406(G)- has been reported to be an eQTL for *HLA-DQB1* (*P* = 1.0 × 10$^{-10}$) and *C4A* (*P* = 1.1 × 10$^{-6}$) in EBV-transformed lymphocytes in the Genotype-Tissue Expression (GTEx) v8 database (Supplementary Fig. 9 and Supplementary Table 10).

Contrary to our hypotheses about global associations of BL with *HLA* variation, our study did not find significant associations of BL with any of *HLA* alleles categorized as rare our dataset or with *HLA* zygosity in individuals. The possible exceptions were inverse

**Table 2 Top HLA alleles and Haplotypes, locus, and amino acid residue hits from analysis of 4,645 Participants (800 with BL) in EMBLEM and Malawi.**

| Category | Variant | Info[a] | Eff. Allele/ Ref. Allele[b] | Combined Frequency[c] | | Meta-analysis[d] | | | Meta-analysis[e] | | |
|---|---|---|---|---|---|---|---|---|---|---|---|
| | | | | Controls | Cases | OR (95% CI) | $P_{meta}$ | $P_{het}$ | OR (95% CI) | $P_{condition}$ | $P_{het}$ |
| HLA allele | DQA1*04:01 | 0.99 | - | 0.08 | 0.12 | 1.61 (1.32, 1.97) | $3.71 \times 10^{-6}$ | 0.92 | 1.29 (1.03, 1.63) | 0.030 | 0.94 |
| Class II haplotype | DRB1*03:02- DQA1*04:01- DQB1*04:02 | - | - | 0.06 | 0.10 | 1.58 (1.25, 1.99) | $1.08 \times 10^{-4}$ | 0.83 | - | - | - |
| SNP | rs2040406 | 0.98 | G/A | 0.28 | 0.36 | 1.43 (1.26, 1.63) | $4.62 \times 10^{-8}$ | 0.84 | 1.68 (1.09, 2.59) | 0.019 | 0.54 |
| Imputed residue | Gln 53 in HLA-DQA1 chain[f] | 0.98 | - | 0.32 | 0.38 | 1.36 (1.20, 1.55) | $2.06 \times 10^{-6}$ | 0.60 | 0.79 (0.52, 1.21) | 0.28 | 0.36 |

AA amino acid residue, HLA human leukocyte antigen, OR odds ratio, CI confidence interval. All statistical tests are 2-sided.
[a] imputation information ($R^2$) was obtained from SNP2HLA.
[b] Effect allele/reference allele.
[c] For locus rs2040406, the numbers indicate frequencies for the effect allele.
[d] Estimates were obtained from generalized linear mixed models (GLMMs), adjusting for sex, age (continuous), Plasmodium falciparum positivity, population substructure using the top 3 principal components (PCs) and genetic relatedness using a genetic relationship matrix. Association statistics were conducted by a fixed-effect meta-analysis across four countries. P values were corrected for multiple comparisons using Bonferroni adjustment ($P < 2.7 \times 10^{-4}$ for 187 classical HLA alleles and $P < 1.1 \times 10^{-6}$ for other 46,350 variants in the HLA region). Cochran's heterogeneity p value was obtained.
[e] The three HLA variants (DQA1*04:01, rs2040406, and Gln-53) were included in one model to determine if effects for one variant were independent of the other two.
[f] Position 53 is polymorphic with three common residues: Q (Glutamine), K (Lysine), and R (Arginine). L (Leucine) is rare in our population. The analysis compared glutamine versus other variants at position 53.

associations between BL and carriage of common HLA-A alleles (OR = 0.86, P = 0.016, Supplementary Table 11) and homozygosity at HLA-C (OR = 0.62, P = 0.004, Supplementary Table 12). However, independent studies are needed to confirm these findings.

## Discussion

Our study using imputed high-resolution HLA data from four countries in Africa supports the hypothesis that BL risk is related to HLA variation. This hypothesis was proposed five decades ago[2], but has been difficult to confirm or refute because of lack of accurate HLA typing data and well-designed studies. We also report associations between BL risk and carriage of HLA-DQA1*04:01 allele and GWAS SNP rs2040406(G) in the HLA-DQA1 region. These HLA associations may be related to effects of HLA on EBV control because the higher risk rs2040406(G) variant for BL is in high LD ($r^2 > 0.8$) with several GWAS SNPs (e.g., rs1064991) known to have pleiotropic effects against EBV humoral immune response[26,27]. Additionally, both rs2040406(G) and HLA-DQA1*04:01 have pleiotropic effects against multiple sclerosis[30,31], which is EBV-linked[32].

Because malaria is the strongest geographical risk factor for BL and the risk of BL increases by 39% per 100 cumulative infections in a child[33], we hypothesized that HLA associations with BL may be related to control of malaria. However, neither HLA-DQA1*04:01 nor rs2040406(G) alleles were not associated with P. falciparum infection detection or parasite density. Our finding of a null association between BL and HLA-B*53 confirms a previous null result from a smaller study[34], and is consistent with the null association between severe malaria and HLA-B*53 reported in a malaria GWAS of 17,000 individuals from 11 countries[11]. These results cast doubt on the frequently cited result of HLA-B*53 as a marker of malaria resistance[9,35].

Several of our findings may suggest that there are other mechanisms that could link HLA variation to risk of BL. For example, HLA-DQA1*04:01 has been linked to autoimmune-related conditions, such as Henoch-Schönlein Purpura[36]. Is it possible that the association with HLA-DQA1*04:01 is pointing to a role of autoimmunity and risk of BL. This inference was suggested by reports of an inverse association between a history of allergy or asthma and BL risk among young adults in the International Lymphoma Consortium study[37,38]. It is also supported by findings that BL tumors display skewed usage of immunoglobulin heavy-chain variable gene segment 4 (IGH-V4), which is implicated in autoreactivity[39].

The results also suggest immuno-genetic mechanisms in the etiology of BL that can be investigated by experimental methods, particularly with regard to the control of EBV infection as a critical factor affecting risk of BL. These include functional studies of expression of cytokines associated with Th1 (IFN-γ), Th2 (IL-10), or Treg (IL-17) phenotypes[40] or the assessment of CD4+ and CD8+ T cell effector memory subsets targeting selected EBV proteins according to HLA type (HLA-DQA1*04:01 versus not). While previous studies have primarily centered on EBNA1[41], future research could be expanded to employ broader panels of EBV proteins, including the 33 EBV proteins, such as the viral capsid antigens (VCAp18, -p23, -p40, -p160), which showed to be differentially reactive in BL cases compared to controls. The studies could also use a smaller set of peptides, such as the four-marker immune panel, including BHRF1 (Bcl-2 homolog), BMRF1 (EAp47), BBLF1 (tegument protein), and BZLF1 (ZTA), whose reactivity most accurately classified BL status in patients in Ghana[42].

Our study was possible because imputation of HLA alleles from GWAS data in sub-Saharan Africa, which we show to be a feasible and a practical way to obtain high-resolution HLA data to investigate the relationship between HLA variation and BL risk.

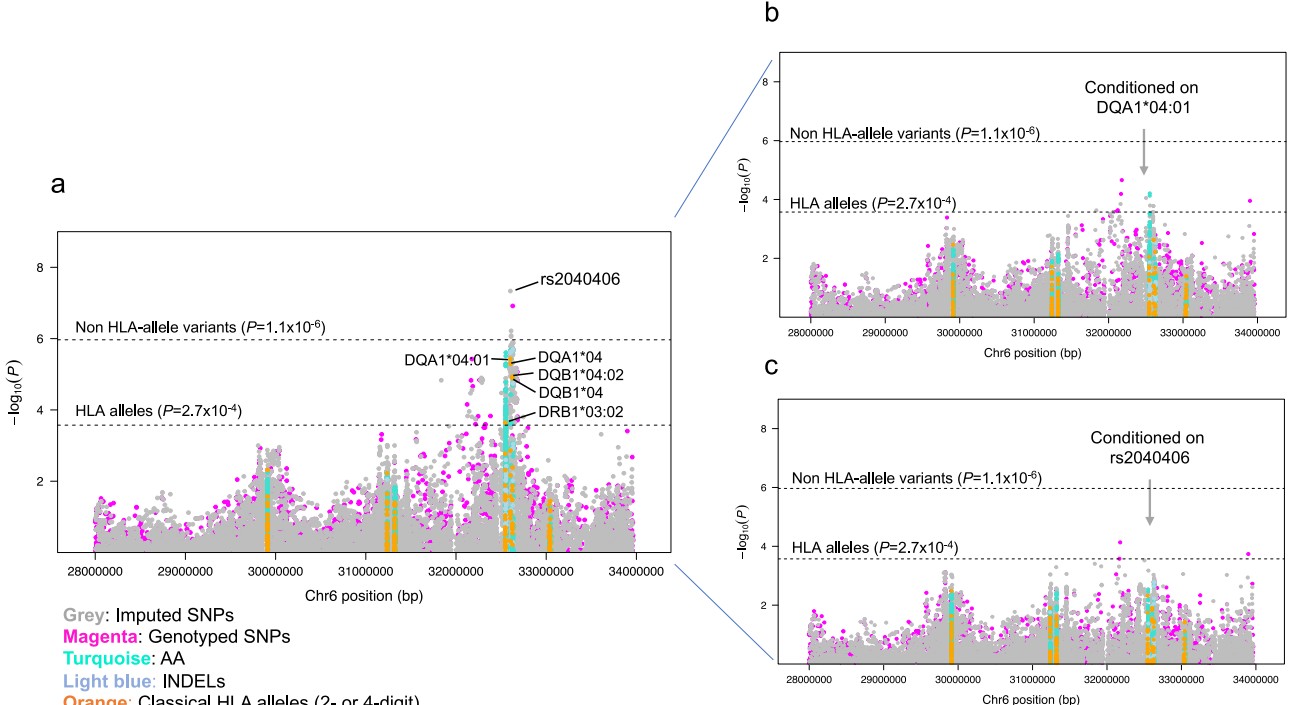

**Fig. 2 Region association plots of variants in the human leukocyte antigen region and Burkitt lymphoma based on 800 BL cases and 3,845 controls.**
**a** Shows results from unconditional analyses; **b** shows analyses conditional on DQA1*04:01; **c** Shows results conditional on rs2040406. The color of the dots indicates different polymorphisms: grey dots indicate imputed SNPs; magenta dots indicate genotyped SNPs; turquoise dots indicate amino acids (AA); light blue dots indicate indels; and orange dots indicate classical HLA alleles at 1- or 2-field resolution.

Our study adds value to the ongoing GWAS and expands the experience with HLA imputation in African subjects[25], who are currently underrepresented in genetic studies in Africa[23].

We note several strengths of our study, including having the largest sample size to date, enrollment from four countries in Africa, availability of SBT data for evaluating the imputation accuracy, and detailed information on covariates to adjust for confounding. However, we acknowledge weaknesses, including lack of anti-EBV antibodies results, which limits our ability to discuss the link between HLA and EBV control. Although our study sample size is the largest to date, it is still relatively small for genome-wide studies and only included people from East Africa. Thus, our results may not be generalizable to other SSA populations. We did not use a GWAS significance threshold (e.g., $5 \times 10^{-8}$), which is conservative for some of our exploratory hypotheses. Larger studies with more broad sampling across different countries in SSA are needed to confirm or refute our findings. Such studies may improve the capacity to investigate HLA in SSA, increase clarity of associations, and identify generalizable results.

To conclude, our findings confirm the high accuracy of HLA imputation from GWAS data in populations with admixed Nilotic ancestry. We report significant association of BL risk with *HLA-DQA1*04:01* and rs2040406(G) in East African children. We hypothesize that the observed associations could be mediated by HLA effects on control of EBV infection and/or autoimmunity, and suggest a promising area of research might be understanding the link between HLA variation and EBV control in SSA populations.

## Methods
**Study population.** The methods of the EMBLEM[24] and the Infections and Childhood Cancer studies have previously been published[43]. The EMBLEM study is a population-based study that enrolled cases and controls aged 0-15 years in two neighboring regions in Uganda and in four neighboring regions in

Kenya and Tanzania (Fig. 1). BL diagnosis was based on local histological/cytological diagnosis (74% of included cases) or clinical, imaging, and laboratory diagnosis. There were no apparent demographic or clinical differences between these groups. The controls were apparently healthy children enrolled from random villages (100 in Uganda, 100 in Kenya, and 95 in Tanzania)[24]. HIV-positive participants in EMBLEM were not excluded because HIV infection was rare (24 BL cases and 15 controls)[24]. In Malawi, participants were children 0–15 years enrolled with cancer at Queen Elizabeth Hospital in Blantyre. BL diagnosis was based on local cytology or histology and compatible clinical investigations; children with non-lymphoid solid cancers were used as controls. Participants with HIV or Kaposi sarcoma were excluded to protect their privacy. Demographic and risk factor information was obtained via a structured interviewer-administered questionnaire[24]. Venous blood was collected in EDTA tubes at enrollment. In EMBLEM, *P. falciparum* infection in blood was determined using microscopy of thick-film blood smears, or antigen capture rapid diagnostic tests (RDTs)[24]. In Malawi, *P. falciparum* infection was based on *P. falciparum*-specific polymerase chain reaction (PCR)[43]. Because we have previously shown distinct ancestry patterns in East versus West African populations[26], we conducted comparative studies of HLA patterns in 968 Ghanaian adult men who were previously studied for ancestry to compare with our East African populations.

**Ethical approvals.** We confirm that all relevant ethical regulations were followed. Specifically, approval for the EMBLEM study was granted by ethics committees at the Uganda Virus Research Institute (GC/127), Uganda National Council for Science and Technology (H816), Tanzania National Institute for Medical Research (NIMR/HQ/R.8c/Vol. IX/1023), Moi University/Moi Teaching and Referral Hospital (000536), and National Cancer Institute (10-C-N133). Ethical approval for the original Infections

and Childhood Cancer Study was granted by ethics committees at the Malawi College of Medicine (P.03/04/277R) and Oxford University. Because the original Malawi Infections and Childhood Cancer study did not request participants to consent to genetic testing, special ethical approval to conduct genetic testing was obtained from the Malawi National Health Sciences Research Committee (Approval #2405). Written informed consent was obtained from participants' guardians in EMBLEM and Malawi studies, and written informed assent was obtained from children aged ≥7 years old in the EMBLEM study. The ethical approval for genetic studies favored research that would enable possible or suitable interventions in the communities where participants were enrolled or increasing knowledge considered relevant to the local communities, such as research on *HLA* variation and malaria resistance to investigate the association of BL with malaria.

**DNA extraction and genotyping procedures**. DNA extraction and genotyping were performed at the Cancer Genomics Research Laboratory, NCI, USA. Genotypes of approximately 4.6 million variants were determined using the Infinium Omni5Exome-4 v1.3 BeadChip (Illumina, San Diego, CA, USA) following standard Illumina data analysis workflow[26]. Genotype data were phased and then imputed using the African Genome Resources panel on the Sanger imputation server (https://imputation.sanger.ac.uk/). This panel was preferred because it currently has the largest number of genomes from African participants, including from the Great Lakes region in Africa. Only variants imputed with high confidence (genotype imputation quality info ≥0.9) with a minor allele frequency (MAF) ≥ 0.01 were retained in the analysis dataset. All variants analyzed passed Hardy–Weinberg equilibrium (HWE) test in the controls using a threshold of $P < 1.0 \times 10^{-6}$. Ancestry was evaluated using principal components (PC) analysis of 787,731 genotyped uncorrelated ($r^2 < 0.3$) SNPs outside the *HLA* region. The top three country-specific PCs were used to control for ancestry. Additionally, because of a high degree of relatedness in Ugandan controls[26], we constructed a genetic relationship matrix (GRM) for all individuals in the dataset, based on the probability that two individuals i and j share 0, 1, or 2 alleles identical by descent (IBD)[44], and used it to control for relatedness.

We performed HLA imputation using Minimac4[45] on the Michigan Imputation Server (MIS) with a multi-ancestry reference panel which contains data from 21,546 unrelated individuals[25]. We applied the default quality control procedures of the MIS pipeline. We imputed eight classical *HLA* genes *HLA-A, B, C* and *HLA-DRB1, DQA1, DQB1, DPA1*, and *DPB1*, and amino acids and intergenic variants using genotypes extracted from chr6:25Mb-35Mb (hg19/GRCh37; $n = 49,159$ SNPs) for the 4645 participants. In total, we inferred 1439 classical I and II alleles at 1- and 2-field resolution, 164 INDELs, and 4,511 amino acid polymorphisms (Supplementary Table 1). After quality control (Supplementary Fig. 10), we retained 187 classical HLA alleles, 126 INDELs, 2652 amino acid polymorphisms, and 43,572 GWAS SNPs (11,702 genotyped and 31,870 imputed) with imputation $R^2 > 0.8$ and MAF > 1% for association analysis. The HLA allele frequencies of East Africa, when compared with Ghana population[34], are presented in Supplementary Fig. 11.

**Statistics and reproducibility**. We assessed the accuracy of our *HLA* imputation by examining the concordance and correlation between the imputed classical *HLA* alleles and SBT-*HLA* genotypes of 600 participants in the same dataset with paired data[22]. We assessed BL and *HLA* allele, SNP, and single amino acid residue associations by fitting generalized linear mixed models (GLMM) with the logit link in country-specific datasets with covariates (sex, age, *P. falciparum* infection status, ancestry as fixed effects, and GRM as a random effect) as the main model. *P. falciparum* is the strongest known co-factor for the geographic patterns of BL[24,33,46,47]. Thus, it was considered a-priori, as a risk factor and a confounder and included in our main models. When interpreted as a confounder, any associations with *HLA* alleles that remain significant indicate that additional contribution of *P. falciparum* was not responsible for the observed associations with *HLA* alleles.

The main results are presented as summary odds ratios (ORs) and significance assessed by 95% confidence intervals (95% CIs), computed using a standard meta-analytical approach of country-specific ORs in PLINK v1.9. *HLA* variants were coded as 0, 1, or 2 corresponding to the number of variant alleles carried by an individual. Formal analysis of statistical heterogeneity of associations with BL across the countries was assessed using Cochran's Q test. *P* values were calculated using Wald tests and corrected for multiple comparisons using Bonferroni adjustment ($P < 2.7 \times 10^{-4}$ for 187 classical HLA alleles and $P < 1.1 \times 10^{-6}$ for another 46,350 variants in the HLA region). While a more conservative *P* value of GWAS significance (e.g., $5 \times 10^{-8}$) may be considered more rigorous, it was not preferred for the current analysis because we wanted to test hypotheses based on prior epidemiological and biological data about *HLA* associations with malaria[10] or EBV[27], or with BL[23]. We performed conditional single-variant association tests, where in addition to the confounders included in the models, we performed conditional adjustment for HLA alleles or SNPs identified in the main models to be significantly associated with BL. Associations of BL with 12 variants with a priori associations with BL, severe malaria or EBV were assessed without correcting for multiple comparisons (Supplementary Table 2). Alleles that are polymorphic at the 2-field resolution are reported as such, while those with limited polymorphism are reported at 1-field resolution.

We examined the robustness of our results by conducting several sensitivity analyses. We excluded *P. falciparum* infection from the main model in sensitivity model 1. In the subsequent sensitivity models 2-5, we controlled for urban vs. rural residence of participants (model 2), wet vs. dry season of enrollment (model 3), carriage of *HBB*-rs334(A) and *ABO*-rs8176703(T) that are also protective against BL[48] (model 4), and excluded 14 individuals with outlier principal components (model 5).

Finally, we defined *HLA* allele groups as common, when the allele frequency [AF] >5%, otherwise as not common when the allele frequency was 1%-5%. Alleles with <1% of the participants were not included in this analysis. This variable was used to investigate global HLA associations of BL with common versus not common *HLA* alleles, which provides a way to assess potential detrimental associations with rare *HLA* alleles, based on the assumption that natural selection influences allelic distribution and favorable alleles will be differentially represented in cases and controls[49]. Second, we investigated associations between BL and *HLA* zygosity at six *HLA* loci (*HLA-A, B, C* and *HLA-DRB1, DQB1*, and *DPB1*). This approach was selected to reduce potential selection bias, as it could arise from excluding individuals carrying rare *HLA* alleles, particularly given their higher likelihood of being heterozygotes. *HLA* zygosity in an individual was defined as the number of alleles at the 2-field resolution that were homozygous across the six HLA loci. We hypothesized that low HLA zygosity (i.e., with more loci being heterozygous, is correlated with a broader, and therefore, more effective immune response repertoires, while high zygosity would be correlated with a decreased repertoire of immune responses[50]. If so, then we reasoned that low zygosity might be associated with decreased BL risk, perhaps, mediated by more effective immune responses against relevant infections, e.g., EBV or malaria.

Because our understanding of associations between classical *HLA* alleles and asymptomatic *P. falciparum* infection, particularly in older children, is less clear, we assessed associations between *HLA* variants with *P. falciparum* infection among controls. We used similar models as those used for analysis of BL, except with infection status as the outcome in the controls. To gain insights about whether *HLA* alleles that were significantly associated with BL have effects on *P. falciparum* infection, we examined the relationship between those alleles with log-transformed *P. falciparum* density in the controls. These analyses were performed separately for EMBLEM and Malawi participants because in Malawi parasite density was measured using PCR, which is more sensitive, while in EMBLEM parasite density was measured using blood smears, which is less sensitive.

Statistical analyses were performed in R (version 4.1.0) utilizing computational resources of the NIH HPC Biowulf cluster.

**Haplotype inference, visualization, and comparisons across countries**. We constructed *HLA* haplotypes separately for each country from phased genotypes using Haplo.stats package v.1.7.7. *HLA* linkage disequilibrium (LD) and/or haplotype configurations across multiallelic genetic markers were visualized using Disentangler plots (http://kumasakanatsuhiko.jp/projects/disentangler). We tested associations between BL with each haplotype, controlling for covariates used in the main model. We also descriptively compare *HLA* patterns in the four countries in East Africa versus in Ghana ($N = 968$), which we previously showed to be ancestrally different from the populations in East Africa[26].

**Reporting summary**. Further information on research design is available in the Nature Portfolio Reporting Summary linked to this article.

## Data availability

The genome-wide and phenotypic data from the Ghana Prostate Healthy Study and the EMBLEM and Malawi studies are publicly available. The previously published data from the Ghana Prostate Healthy Study are available under restricted access in the Genomic Data Commons; access to these studies is available through dbGaP: Ghana Prostate Study (phs000838.v1.p1). The genetic data of participants in the EMBLEM and Malawi studies are under controlled access (requires IRB approval and are limited to not-for-profit research). Readers can access the data by applying via dbGaP under accession link phs001705.v1.p1. The source data behind the graphs in Fig. 2 are available in Supplementary Data 1.

## Code availability

All code was written in R, or Bash. Scripts, required software packages, and instructions are available on the Github repository (https://github.com/smbulaiteye/HLA-EMBLEM.git)[51].

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

## Acknowledgements

We thank the study population and communities for their participation. We thank Ms. Janet Lawler-Heavner at Westat Inc, (Rockville, MD, USA) and Mr. Erisa Sunday at the African Field Epidemiology Network (Kampala, Uganda) for managing the study. We are grateful to the leadership of the collaborating countries and institutions for hosting local field offices and laboratories and supporting the fieldwork. We thank Ms. Laurie Buck, Dr. Carol Giffen, and Mr. Greg Rydzak at Information Management Services Inc. (Calverton, MD, USA) for coordinating data, and preparing data analysis files. The study was funded by the Intramural Research Program of the Division of Cancer Epidemiology and Genetics, National Cancer Institute (NCI) (Contracts HHSN261201100063C and HHSN261201100007I), and the Intramural Research Program, National Institute of Allergy and Infectious Diseases (SJR), National Human Genome Research Institute (MHG), National Institutes of Health (NIH), Department of Health and Human Services. The authors acknowledge the research contributions of the Cancer Genomics Research Laboratory for their expertise, execution, and support of this research in the areas of project planning, wet laboratory processing of specimens, and bioinformatics analysis of generated data from the NCI, NIH, under NCI Contract No. 75N910D00024. Y.L. is supported by a Kennedy Trust KTRR Senior Research Fellowship (KENN202109). The content of this publication does not necessarily reflect the views or policies of the Department of Health and Human Services, nor does mention of trade names, commercial products, or organizations imply endorsement by the US Government. The content of this manuscript is the sole responsibility of the authors. The funders had no role in study design, data collection, analysis or interpretation, or writing and submitting the report for publication.

## Author contributions

Z.L. and S.M.M. wrote the manuscript. S.M.M., M.D.O., S.J.R., P.K. conceived the idea, designed the EMBLEM study and supervised fieldwork. S.M.M., O.O.O., S.R. and Y.L. conceived the idea of conducting *HLA* imputation. S.M.M., S.K., R.B. conceived the idea of SBT *HLA* typing. M.D.O., P.K., G.C., W.T.J., E.B., S.J.R., C.N.T., P.A.W., R.T.K., W.N.W., N.M., E.K., T.K., I.O., I.D.L., H.N., H.D., L.W.A., K.B., J.J.G., G.N.L., S.K., N.M., E.M.M., R.N., A.W.H., J.E.M., A.A.A. and C.M. conducted, monitored or advised on field work. B.H., N.C., J.L., M.M., W.L., M.Y., A.H., R.B., M.V., S.K. conducted genetic testing and processed genetic data. M.H.G., Z.L., Y.L., S.R. analyzed genetic data and developed primary and sensitivity models; L.P.O. and M.C. advised on HLA genetic analyses; Y.L. and S.R. performed HLA imputation; S.J.C. provided expertise on genetic analysis, and interpretation of data. All authors read and approved the final manuscript.

## Competing interests

The authors declare no competing interests.

## Additional information

[1]Division of Cancer Epidemiology and Genetics, National Cancer Institute, Rockville, MD, USA. [2]Kennedy Institute of Rheumatology, Nuffield Department of Orthopaedics, Rheumatology and Musculoskeletal Sciences, University of Oxford, Oxford, UK. [3]Center for Data Sciences, Brigham and Women's Hospital, Harvard Medical School, Boston, MA, USA. [4]Division of Rheumatology, Immunology, and Immunity, Brigham and Women's Hospital, Harvard Medical School, Boston, MA, USA. [5]Division of Genetics, Brigham and Women's Hospital, Harvard Medical School, Boston, MA, USA. [6]Department of Biomedical Informatics, Harvard Medical School, Boston, MA, USA. [7]Broad Institute of MIT and Harvard, Cambridge, MA, USA. [8]College of Health Sciences, Makerere University, Kampala, Uganda. [9]Institute of Transfusion Medicine and Transplant Engineering, Hanover, Germany. [10]Center for Research on Genomics & Global Health, NHGRI, National Institutes of Health, Bethesda, MD, USA. [11]St. Mary's Hospital, Lacor, Gulu, Uganda. [12]EMBLEM Study, African Field Epidemiology Network, Kampala, Uganda. [13]Kuluva Hospital,

Arua, Uganda. [14]Division of Intramural Research, National Institute of Allergy and Infectious Diseases, National Institutes of Health, Bethesda, MD, USA. [15]Moi University College of Health Sciences, Eldoret, Kenya. [16]Academic Model Providing Access To Healthcare (AMPATH), Eldoret, Kenya. [17]Bugando Medical Center, Mwanza, Tanzania. [18]Shirati Health, Education, and Development Foundation, Shirati, Tanzania. [19]Department of Pathology, The Ohio State University, Columbus, OH, USA. [20]Cancer Genomics Research Laboratory, Frederick National Laboratory for Cancer Research, Frederick, MD, USA. [21]Departments of Pediatrics and Surgery, Kamuzu University of Health Sciences (formerly College of Medicine), University of Malawi, Blantyre, Malawi. [22]Epidemiology and Cancer Statistics Group, Department of Health Sciences, University of York, York, UK. [23]Cancer Epidemiology Unit, University of Oxford, Oxford, UK. [24]National Health Sciences Research Committee, Research Department, Ministry of Health, Lilongwe, Malawi. [25]Stanford Cancer Institute, Stanford University, Stanford, CA, USA. [26]University of Ghana Medical School, Accra, Ghana. [27]Basic Science Program, Frederick National Laboratory for Cancer Research, National Cancer Institute, Frederick, MD, USA and Laboratory of Integrative Cancer Immunology, Center for Cancer Research, National Cancer Institute, Bethesda, MD, USA. [28]Ragon Institute of MGH, MIT and Harvard, Cambridge, MA, USA. [29]These authors contributed equally: Zhiwei Liu, Yang Luo. [30]These authors jointly supervised this work: Soumya Raychaudhuri, Sam M. Mbulaiteye. ✉email: mbulaits@mail.nih.gov

