## [Peer Review File · Communications Biology]

Reviewers' comments:

Reviewer #1 (Remarks to the Author):

The authors present an HLA association study for Burkitt lymphoma in sub-Saharan African children. The sample size of cases and controls is large and SNP genotyping is well described. The HLA alleles were imputed from SNPs and targeted HLA genotyping was used to validate accuracy of this process. The results could benefit more information on linkage between HLA alleles and SNPs. The relationships between the HLA alleles, the HLA amino acid residues, and the SNPs could be much better described.

I have the following comments on this manuscript:

- Introduction - Provide some quantification of how much higher BL incidence is in SSA vs the rest of the world.
- Introduction - Can you explain further why elevated anti-EBV antibodies would indicate poor EBV control? Wouldn't protective HLA Class II associations lead more readily to development of anti-EBV antibodies by presenting peptides to CD4+ T cells and providing help to B cells to make antibodies? HLA Class I responses would pre result in direct killing by CD8+ T cells.
- Introduction - Indicate that the SBT typing of HLA alleles in the Kirimunda et al. study was "in a previous study". I didn't realize you weren't talking about the present until later.
- Methods - Indicate the algorithm used to impute the HLA alleles from SNPs in the main text - looks like it was SNP2HLA, but that's indicated only in the figures? Why did you choose SNP2HLA instead of HIBAG?
- Results - Page 12 - Lines 293-294. Can you provide linkage disequilibrium values between the DQA1, DQB1, and DRB1 alleles that had elevated risk based control haplotype data? What are the common haplotypes containing DQA1*04:01 in each population? Were there any Class II haplotypes that contained DQA1*04:01 besides DRB1*03:02-DQA1*04:01~DQB1*04:02 that were significant? Indicate the population frequency of DQA1*04 alleles that aren't DQA1*04:01.
- Results Page 12 - Lines 305-307 - It seems unnecessary to test if haplotype association would hold up after conditioning on one of the alleles contained in the haplotype. There's almost no conceivable way the haplotype association would remain significant. Remove this result.
- Results - Lines 317-322 - Please indicate if Gln53 is present in DQA1*04:01. If you test for associations of Gln53-containing DQA1 alleles excluding DQA1*04:01, do you find any significant associations? What are the other frequent Gln53-containing DQA1 alleles in these populations? I'm looking for a reason to believe this is a useful independent finding.
- Results - What is the LD between DQA1*04:01 and SNP rs2040406? Can you test if cases that have both variants are at even higher risk?
- Results - rs2040406 is indicated as an eQTL for HLA-DQB1. Please tell us if the higher risk variant is associated with increased or decreased expression of HLA-DQB1. Can you also highlight this fact in the abstract?
- Discussion - Can you describe in more detail what future mechanistic studies would aim to do next with this information?

Minor issues:

- Abstract - Typo - Line 94 - "HLA alleles wth frequency" should be "HLA alleles with frequency"
- Abstract - Line 99 - "other allelic variant" should be "other amino acid variants".
- Abstract - Typo - Line 101 - "singficaint" should be "significant"
- Methods - Page 9 - Line 221 - "2-filed" should be "2-field"
- Methods - Page 9 - Line 228 - "outlier principal component" should be "outlier principal components"
- Methods - Page 9 - Link 237 - "wold be correlated" should be "would be correlated"
- Results - Page 14 - Typing "if cance can be excluded" should be "if cancer can be excluded"

- Supplementary Figure 4 - "B53:01" should be B*53:01. "since no 4-digit HLA alleles" should be "since there were no 4-digit HLA alleles".

Reviewer #2 (Remarks to the Author):

Liu et al., explored associations between HLA variants and the risk to develop Burkitt lymphoma in African population using a multi-center design. They reported independent associations at HLA-DQA1*04:01, rs2040406 and the 53 amino acid position of HLA-DQA1 to be associated with BL. The statistical analyses are straightforward. Below are some concerns to address.

1. Since the samples were genotyped with genome-wide arrays, are there any non-HLA loci associated with BL?

2. The authors defined two statistical significance thresholds for HLA alleles and HLA variants using Bonferroni correction. Should a more conservative P value be used, like $5e-8$ of GWAS significance? Moreover, the associations are not independently replicated, though it might be challenging to collect large BL samples for African population. This limitation should be least discussed in the discussion section. In addition, as there are different ways of P-adjustment, it would be clearer to the readers by indicating which P value was adjusted or not in the text and table.

3. Serological EBNA1 antibodies are implicated with re-/activation of EBV and certain types of malignancy. Not all SNPs associated with anti-EBV VCA IgG antibodies in southwest Uganda were associated with BL in individuals of African ancestry. Any association with BL for the EBNA1-antibody-related SNPs in other non-African population? It would be more informative to expand these in the result and discussion.

4. It's a bit confusing for the methods that the authors employed to gain insights about whether BL associated HLA alleles have effects on *P. falciparum* infection. First, the authors focused on analyses with control samples. The authors claimed no associations of HLA alleles with *P. falciparum* in controls (Fig S9 and S10), but the pathogen infection seems a protective factor for BL (35.3% in cases vs 48.3% in controls). Moreover, it's unclear why the infection was adjusted in the GLMMs reported in Table 2. Any associations remained significant in this joint model would indicate additional contributors to BL risk other than the infection.

5. The author used "broad groupings of HLA alleles as common (allele frequency [AF] >5%) otherwise as rare to investigate global HLA associations with BL." According to their QC procedure, which removed variants with MAFs lower than 1%, the "rare" category consisted of variants with MAFs between 1%-5%. With such a range of MAFs, the variants are unlikely considered as rare.

6. Was the definition of HLA zygosity taking into account the frequencies of HLA alleles?

7. Additionally, there are some typos in the manuscript, like "singficaint" in line 101 and "cance" in line 338.

Reviewers' comments:

Reviewer #1 (Remarks to the Author):

The authors present an HLA association study for Burkitt lymphoma in sub-Saharan African children. The sample size of cases and controls is large and SNP genotyping is well described. The HLA alleles were imputed from SNPs and targeted HLA genotyping was used to validate accuracy of this process. The results could benefit more information on linkage between HLA alleles and SNPs. The relationships between the HLA alleles, the HLA amino acid residues, and the SNPs could be much better described.

I have the following comments on this manuscript:

- Introduction - Provide some quantification of how much higher BL incidence is in SSA vs the rest of the world.

RESPONSE: We thank the reviewer for their comments. As suggested by the reviewer, the BL incidence information has been added to the Introduction section in the 1st paragraph on page 4, which is shown below:

“BL incidence varies 3- to 5-fold within and across continents, but the highest incidence is recorded in children in sub-Saharan Africa (SSA, 2-4 per 100,000 person-years [ref. #3-#4]), where it is approximately 10-fold higher than the incidence in the United States or Europe [ref. #1 and #5].”

- Introduction - Can you explain further why elevated anti-EBV antibodies would indicate poor EBV control? Wouldn't protective HLA Class II associations lead more readily to development of anti-EBV antibodies by presenting peptides to CD4+ T cells and providing help to B cells to make antibodies? HLA Class I responses would pre result in direct killing by CD8+ T cells.

RESPONSE: We appreciate the reviewer's insight on this issue. We agree that high antibody levels may not necessarily indicate poor EBV control, i.e., an inability to control EBV infection. High anti-EBV antibodies are usually interpreted as indicating exposure to a high EBV burden, where antibody levels correlate with current or cumulative viral burden (Maurmann S et al., 2003; Besson C et al., 2006). Because EBV infection involves latent infection, during which EBV remains silent in B cells, and lytic infection, when infected cells (B or epithelial cells) produce virions that are shed into circulation and saliva, recurrent lytic infection, such as occurs following recurrent malaria episodes (Lam KM et al., 1991), is one mechanism that leads to exposure to a high viral burden in people who are otherwise able to control the infection. We agree with the reviewer that the immune response to EBV is modulated by CD4+ helper 2 (Th2) cells providing help to antibody producing B cells in generating anti-EBV antibodies [Kidd P, et al. 2003; Mosmann TR et al., 1989; Paul WE et al., 2010; Zhu J et al., 2008]. Thus, high anti-EBV-antibodies may indicate a robust Th2 response to control EBV viral load and titers correlate with current or cumulative load (Maurmann S et al., 2003; Besson C et al., 2006). It is worth noting that a Th2-type response might also be associated with tolerance of innocuous levels of antigenemia (Koyasu S et al., 2011). Conversely, a stronger Th1 response, involving CD8+ T helper cells involves direct killing, and is frequently associated with severe EBV symptomatic diseases, such as chronic active EBV infection or infectious mononucleosis (Liu M et al., 2023). To clarify, we have revised the sentence in the 1st paragraph on page 4 as follows.

“HLA variation has also been associated with EBV control, including *HLA-A*02:01* [ref. #13] and *HLA-DQB1*02* [ref. #14] which are associated with elevated anti-EBV antibodies. *HLA-A*02:01* is part of the HLA Class I system and may mediate direct killing of EBV-infected cells by modulating expression of CD8+

Th1-type immune response (ref. #15). Conversely, HLA-DQB1*02 is part of the HLA Class II system and may mediate EBV control by facilitating anti-EBV antibody secretion through the modulation of peptide presentation to CD4+ T cells and their Th-2 signaling to B cells, thereby promoting an antibody response (ref. #16, #17). Therefore, the reported HLA associations are compatible with control of immune response to EBV or malaria as potential mechanisms for influencing risk of BL.”

Reference list:

Maurmann S, Fricke L, Wagner HJ, Schlenke P, Hennig H, Steinhoff J, Jabs WJ. Molecular parameters for precise diagnosis of asymptomatic Epstein-Barr virus reactivation in healthy carriers. *J Clin Microbiol.* 2003 Dec;41(12):5419-28. doi: 10.1128/JCM.41.12.5419-5428.2003. PMID: 14662920; PMCID: PMC308959.

Besson C, Amiel C, Le-Pendeven C, Brice P, Fermé C, Carde P, Hermine O, Raphael M, Abel L, Nicolas JC. Positive correlation between Epstein-Barr virus viral load and anti-viral capsid immunoglobulin G titers determined for Hodgkin's lymphoma patients and their relatives. *J Clin Microbiol.* 2006 Jan;44(1):47-50. doi: 10.1128/JCM.44.1.47-50.2006. PMID: 16390946; PMCID: PMC1351946.

Lam KM, Syed N, Whittle H, Crawford DH. Circulating Epstein-Barr virus-carrying B cells in acute malaria. *Lancet.* 1991 Apr 13;337(8746):876-8. doi: 10.1016/0140-6736(91)90203-2. PMID: 1672968.

Kidd P. Th1/Th2 balance: the hypothesis, its limitations, and implications for health and disease. *Altern Med Rev.* 2003 Aug;8(3):223-46. PMID: 12946237.

Mosmann TR, Coffman RL. TH1 and TH2 cells: different patterns of lymphokine secretion lead to different functional properties. *Annu Rev Immunol.* 1989;7:145-73. doi: 10.1146/annurev.iy.07.040189.001045. PMID: 2523712.

Paul WE, Zhu J. How are T(H)2-type immune responses initiated and amplified? *Nat Rev Immunol.* 2010 Apr;10(4):225-35. doi: 10.1038/nri2735. PMID: 20336151; PMCID: PMC3496776.

Zhu J, Paul WE. CD4 T cells: fates, functions, and faults. *Blood.* 2008 Sep 1;112(5):1557-69. doi: 10.1182/blood-2008-05-078154. PMID: 18725574; PMCID: PMC2518872.

Koyasu S, Moro K. Type 2 innate immune responses and the natural helper cell. *Immunology.* 2011 Apr;132(4):475-81. doi: 10.1111/j.1365-2567.2011.03413.x. Epub 2011 Feb 16. PMID: 21323663; PMCID: PMC3075501.

Liu M, Wang R, Xie Z. T cell-mediated immunity during Epstein-Barr virus infections in children. *Infect Genet Evol.* 2023 Aug;112:105443. doi: 10.1016/j.meegid.2023.105443. Epub 2023 May 16. PMID: 37201619.

- Introduction - Indicate that the SBT typing of HLA alleles in the Kirimunda et al. study was "in a previous study". I didn't realize you weren't talking about the present until later.

RESPONSE: We thank the reviewer for pointing out the confusing way we introduced the SBT study Kirimunda et al. We have revised the sentence accordingly on page 5 as follows.

“Only one study, which was conducted by our group previously, has used sequence-based typing (SBT) (ref. #23) to obtain accurate high-resolution data (≥ 2 fields) in 600 participants (including 200 with BL) in the Epidemiology of Burkitt Lymphoma in East African Children and Minors (EMBLEM) study in Uganda. (ref. #24)”

- Methods - Indicate the algorithm used to impute the HLA alleles from SNPs in the main text - looks like it was SNP2HLA, but that's indicated only in the figures? Why did you choose SNP2HLA instead of HIBAG?

RESPONSE: We thank the reviewer for this comment and acknowledge that our original text did not make it clear that we used Minimac4 on the Michigan Imputation Server (MIS) for HLA imputation. We have revised our manuscript for clarity.

We recognize that there are several programs available for HLA imputation, all with similar performance levels. We note that the key factor influencing imputation accuracy is the HLA reference panel itself [<https://doi.org/10.1371/journal.pone.0291437>]. The inclusion of individuals from the same populations of interest in the reference panel is crucial for imputation performance. To ensure a comprehensive reference, we opted for a multi-ancestry reference panel consisting of 21,546 unrelated global individuals, including 7,849 individuals with African genetic ancestry. To our knowledge, this panel is the largest panel that covers global and African populations to date. This panel is publicly available on the MIS, but not released as pre-fit classifier, as required in HIBAG imputations. We are therefore unable to use and benchmark HIBAG in our study.

We improved our description of HLA imputation in the revised manuscript:

“We performed HLA imputation using Minimac4 (ref. #30) on the Michigan Imputation Server (MIS) with a multi-ancestry reference panel which contains data from 21,546 unrelated individuals (ref. #26). We applied the default quality control procedures of the MIS pipeline. We imputed eight classical HLA genes HLA - A, B, C and HLA - DRB1, DQA1, DQB1, DPA1, and DPB1, and amino acids and intergenic variants using genotypes extracted from chr6:25Mb-35Mb (hg19/GRCh37; n=49,159 SNPs) for the 4,645 participants.”

- Results - Page 12 - Lines 293-294. Can you provide linkage disequilibrium values between the DQA1, DQB1, and DRB1 alleles that had elevated risk based control haplotype data? What are the common haplotypes containing DQA1*04:01 in each population? Were there any Class II haplotypes that contained DQA1*04:01 besides DRB1*03:02-DQA1*04:01~DQB1*04:02 that were significant? Indicate the population frequency of DQA1*04 alleles that aren't DQA1*04:01.

RESPONSE: We appreciate the request from the reviewer. The linkage and haplotype patterns were previously included in Supplementary Figure 6.

We used a threshold of 1% to analyze haplotypes. There were only two haplotypes that met this threshold, namely, DRB1*03:02-DQA1*04:01-DQB1*04:02 and DRB1*08:04-DQB1*03:01-DQA1*04:01, with a frequency of 7% and 1%, respectively. DRB1*08:04-DQB1*03:01-DQA1*04:01 was not associated with BL (OR=1.44, 95%CI=0.82-2.51; p=0.200). We have revised the manuscript accordingly.

Similarly, only one other DQA1*04 allele other than DQA1*04:01 was observed, namely, HLA-DQA1*04:03N, but it was rare (0.04%) and was filtered based on our threshold for analysis. Our changes to the main text are shown below.

“For example, HLA haplotype *DRB1*08:04-DQB1*03:01-DQA1*04:01* was more frequent in Ghana (5%; 45/923) compared to East Africa (1%; 32/4645). This haplotype was not associated with BL risk (OR=1.44, 95% CI=0.82-2.51; p=0.200), whereas *DRB1*03:02-DQB1*04:02-DQA1*04:01*, which was also more frequent in Ghana (10.0%), was associated with elevated BL risk (OR =1.58, 95% CI=1.25-1.99; $P=1.08 \times 10^{-4}$) (**Table 2**)”

- Results Page 12 - Lines 305-307 - It seems unnecessary to test if haplotype association would hold up after conditioning on one of the alleles contained in the haplotype. There's almost no conceivable way the haplotype association would remain significant. Remove this result.

RESPONSE: We appreciate the reviewer's advice. We have deleted the results after conditioning because they are redundant.

- Results - Lines 317-322 - Please indicate if Gln53 is present in DQA1*04:01. If you test for associations of Gln53-containing DQA1 alleles excluding DQA1*04:01, do you find any significant associations? What are the other frequent Gln53-containing DQA1 alleles in these populations? I'm looking for a reason to believe this is a useful independent finding.

RESPONSE: We appreciate the reviewer's probe into whether Gln53 is an independent finding. Gln53 is contained in DQA1*05:01 as well and shows an association with BL (OR=1.17, p=0.027). While the statistical independence of this association is uncertain, considering a type-1 error, the association is of BL with Gln53 is observed both with DQA1*04:01 allele and DQA1*05:01, which consistent with possible biological effect. Because reporting of results should balance both type-1 errors and type-2 errors, we believe the Gln53 result is worth reporting so that future studies can confirm or refute the result.

- Results - What is the LD between DQA1*04:01 and SNP rs2040406? Can you test if cases that have both variants are at even higher risk?

RESPONSE: The LD between *HLA-DQA1*04:01* and SNP rs2040406 was 0.225. The proportion of cases homozygous for both *DQA1*04:01* and rs2040406(GG) was 1.3% and it was 0.68% in the controls. Compared with those not carrying both alleles, those homozygous for both had an OR of 2.66 (95%CI=1.19-5.95; P=0.017). We have revised the manuscript accordingly in the 1st paragraph on Page 14, as shown below:

"Finally, we examined the combined effect of the presence of two risk alleles. Although homozygosity for both *HLA-DQA1*04:01* and SNP rs2040406 (with LD $R^2=0.225$) was low (1.3% in the cases and 0.68% in the controls), children who were homozygous for both had a higher risk for BL (OR=2.66, 95%CI=1.19-5.95; P=0.017) versus those not carrying either allele."

- Results - rs2040406 is indicated as an eQTL for HLA-DQB1. Please tell us if the higher risk variant is associated with increased or decreased expression of HLA-DQB1. Can you also highlight this fact in the abstract?

RESPONSE: As shown in Supplementary Figure 11, the BL risk allele rs2040406-G is associated with decreased expression of HLA-DQB1. Based on the dogma Th2 response dogma discussed above (please refer to response to question #2), it is possible that decreased expression results in decreased presentation of EBV peptides to CD4+ T cells leading to inability to suppress EBV viral load and consequently elevated BL risk as observed. We have highlighted this result in the Abstract following the reviewer's suggestion. The revision is as shown below.

"The higher risk rs2040406(G) variant for BL is associated with decreased *HLA-DQB1* expression in eQTLs in EBV transformed lymphocytes."

- Discussion - Can you describe in more detail what future mechanistic studies would aim to do next with this information?

RESPONSE: We appreciate the opportunity to suggest hypotheses that can be tested in mechanistic studies. Our findings point to immune-genetic mechanisms in the etiology of BL that we posit act by

impairing control of EBV viral activity, thereby permitting oncogenic effects of EBV to influence BL development. Functional studies of cytokines to measure expression of cytokines classically associated with Th1 (FN- γ) or Th2 (IL-10) or Treg (IL-17) phenotype and CD4+ and CD8+ T cell effector memory subsets specific to selected EBV proteins to assess responses by HLA status (HLA**DQA1**04:01) in cases and controls (Szabo SJ et al., 2003). While previous studies have focused on EBNA1, we would propose future research use a larger panel of EBV proteins, including 33 EBV proteins, including the viral capsid antigens (VCAp18, -p23, -p40, -p160) that show differential antibody reactivity in BL cases compared to controls and include four-marker immune signature BHRF1 (Bcl-2 homolog), BMRF1 (EA_{p47}), BBLF1 (tegument protein), and BZLF1 (ZTA) that most accurately classified BL status (Coghill AE et al., 2020).

Reference list:

Szabo SJ, Sullivan BM, Peng SL, Glimcher LH. Molecular mechanisms regulating Th1 immune responses. *Annu Rev Immunol.* 2003;21:713-58. doi: 10.1146/annurev.immunol.21.120601.140942. Epub 2001 Dec 19. PMID: 12500979.

Coghill AE, Proietti C, Liu Z, Krause L, Bethony J, Prokunina-Olsson L, Obajemu A, Nkrumah F, Biggar RJ, Bhatia K, Hildesheim A, Doolan DL, Mbulaiteye SM. The Association between the Comprehensive Epstein-Barr Virus Serologic Profile and Endemic Burkitt Lymphoma. *Cancer Epidemiol Biomarkers Prev.* 2020 Jan;29(1):57-62. doi: 10.1158/1055-9965.EPI-19-0551. Epub 2019 Oct 16. PMID: 31619404; PMCID: PMC6954331.

We have added this information in the Discussion section on Page 16, as follows:

“The results also suggest immune-genetic mechanisms in the etiology of BL that can be investigated by experimental methods, particularly with regard to the control of EBV infection as a critical factor affecting risk of BL. These include functional studies of expression of cytokines associated with Th1 (IFN- γ), Th2 (IL-10), or Treg (IL-17) phenotypes (ref. #49), or the assessment of CD4+ and CD8+ T cell effector memory subsets targeting selected EBV proteins according to HLA type (HLA-DQA1*04:01 versus not). While previous studies have primarily centered on EBNA1 (ref. #50), future research could be expanded to employ broader panels of EBV proteins, including the 33 EBV proteins, such as the viral capsid antigens (VCA_{p18}, -p23, -p40, -p160), which showed to be differentially reactive in BL cases compared to controls. The studies could also use a smaller set of peptides, such as the four-marker immune panel, including BHRF1 (Bcl-2 homolog), BMRF1 (EA_{p47}), BBLF1 (tegument protein), and BZLF1 (ZTA), whose reactivity most accurately classified BL status in patients in Ghana. (ref. #51)”

Minor issues:

- Abstract - Typo - Line 94 - "HLA alleles wth frequency" should be "HLA alleles with frequency"
- Abstract - Line 99 - "other allelic variant" should be "other amino acid variants".
- Abstract - Typo - Line 101 - "singficaint" should be "significant"
- Methods - Page 9 - Line 221 - "2-filed" should be "2-field"
- Methods - Page 9 - Line 228 - "outlier principal component" should be "outlier principal components"
- Methods - Page 9 - Link 237 - "wold be correlated" should be "would be correlated"
- Results - Page 14 - Typing "if cance can be excluded" should be "if cancer can be excluded"
- Supplementary Figure 4 - "B53:01" should be B*53:01. "since no 4-digit HLA alleles" should be "since there were no 4-digit HLA alleles".

RESPONSE: We thank the reviewer for taking the time to review our work and helping correct those typos. These issues have been addressed in the revised version. We have also carefully proofread our manuscript and corrected other typos.

Reviewer #2 (Remarks to the Author):

Liu et al., explored associations between HLA variants and the risk to develop Burkitt lymphoma in African population using a multi-center design. They reported independent associations at HLA-DQA1*04:01, rs2040406 and the 53 amino acid position of HLA-DQA1 to be associated with BL. The statistical analyses are straightforward. Below are some concerns to address.

1. Since the samples were genotyped with genome-wide arrays, are there any non-HLA loci associated with BL?

RESPONSE: We appreciate the reviewer's question. There are non-HLA loci that are associated with BL in the two analyses that have been conducted. The first analysis focused on loci associated with malaria resistance based on prior data indicating that malaria is associated with BL. In those (*a priori*-based analyses), BL was inversely associated with two loci, rs334, which is linked to sickle cell status, and rs8176703, which is linked to blood group O (Blood (2022) 140 (Supplement 1): 9314–9316. <https://doi.org/10.1182/blood-2022-170410>). Because those analyses were done contemporaneously with the HLA analysis, we controlled for those alleles in our current study. The second set of analyses is agnostic using GWAS methods. Two loci in Chr 21q and Chr 6q were significantly associated with BL. These results are still preliminary and additional studies are being conducted to verify them and will be reported in a separate report.

2. The authors defined two statistical significance thresholds for HLA alleles and HLA variants using Bonferroni correction. Should a more conservative P value be used, like $5e-8$ of GWAS significance? Moreover, the associations are not independently replicated, though it might be challenging to collect large BL samples for African population. This limitation should be least discussed in the discussion section. In addition, as there are different ways of P-adjustment, it would be clearer to the readers by indicating which P value was adjusted or not in the text and table.

RESPONSE: We thank the reviewer for these observations. Indeed, we applied two different P-value thresholds to assess statistical significance. We used a nominal threshold of $p < 0.05$ for a small set of HLA alleles that have been previously linked to malaria, EBV, or BL based on the assumption that those alleles would affect BL risk via pathways linked to malaria or EBV, or that our results would replicate the prior associations with BL. For the rest of the alleles, we use a Bonferroni correction based on the actual number of tests that were performed. We considered, as the reviewer suggests, using more conservative P value, like 5×10^{-8} of GWAS significance, but such an approach ignores the wealth of knowledge already known about the etiology of BL, namely association with malaria, EBV, or BL. Such a conservative approach would increase the type-II error, while we agree that our approach appears to increase the type-1 error.

We have revised the manuscript in the Statistical analysis section on Page 9 and result tables to accurately report the P-value thresholds used for each result and why. We also have updated the discussion to explicitly state the limitations of our approach in the Discussion section on Page 17, as follows:

“While a more conservative P value of GWAS significance (e.g., 5×10^{-8}) may be considered more rigorous, it was not preferred for the current analysis because we wanted to test hypotheses based on prior epidemiological and biological data about HLA associations with malaria (ref. #11) or EBV (ref. #34), or with BL. (ref. #24)”

“We did not use a GWAS significance threshold (e.g., 5×10^{-8}), which is conservative for some of our exploratory hypotheses.”

We acknowledge the reviewer’s point that external replication is also crucial. Nevertheless, obtaining large case and control samples for African populations poses significant challenges. This information has been addressed in the Discussion section on Page 17, as shown below.

“Larger studies with more broad sampling across different countries in SSA are needed to confirm or refute our findings. Such studies may improve the capacity to investigate HLA in SSA, increase clarity of associations, and identify generalizable results.”

3. Serological EBNA1 antibodies are implicated with re-/activation of EBV and certain types of malignancy. Not all SNPs associated with anti-EBV VCA IgG antibodies in southwest Uganda were associated with BL in individuals of African ancestry. Any association with BL for the EBNA1-antibody-related SNPs in other non-African population? It would be more informative to expand these in the result and discussion.

RESPONSE: We appreciate the reviewer’s insights. Indeed, we are interested in linking our genetic findings with EBV antibody patterns. The discrepancies noted by the reviewer could be attributed to several factors, including differences in ancestry in the population in Uganda where the EBV study versus the populations where the BL study was done. The Ugandan population, while being African, is much less diverse than the populations contributing to the BL study. It is also possible that non-EBV-related factors, such as malaria or nutrition, might confound these associations. Finally, the results might be limited by power to detect significant findings. The reviewer also raises an interesting question: whether SNPs associated with EBNA1-antibody in non-African population might be used as priors for association with BL. A major concern about this hypothesis is the differences in EBV epidemiology in African (infection during infancy: Piriou E et al., 2012 and usually asymptomatic: Biggar RJ et al. 1978) versus in non-African populations (where more than half of infections occur in young adults [Brodsky AL et al., 1972; Sawyer RN et al., 1971] and is usually asymptomatic), which complicates interpretation of SNP association with EBV in different environments. A second concern is population variation of the identified SNPs across populations. For example, of four SNPs linked with anti-EBNA1-antibody titers in non-African populations (rs2516049 and rs9269233 in Europeans (Zhou Y et al., 2016; Kachuri L et al., 2020), rs477515 and rs2854275 in Mexican American (Rubicz R et al., 2013), one rs9269233 did not show variation in our population, while the other three were unrelated to BL in our study (all p values > 0.05). Overall, our data are sufficient to raise the question about SNPs whose effects may be mediated by EBV control but inadequate to fully identify and elucidate their effects.

Reference list:

Piriou E, Asito AS, Sumba PO, Fiore N, Middeldorp JM, Moormann AM, Ploutz-Snyder R, Rochford R. Early age at time of primary Epstein-Barr virus infection results in poorly controlled viral infection in infants from Western Kenya: clues to the etiology of endemic Burkitt lymphoma. *J Infect Dis*. 2012 Mar 15;205(6):906-13. doi: 10.1093/infdis/jir872. Epub 2012 Feb 1. PMID: 22301635; PMCID: PMC3282570.

Biggar RJ, Henle G, Böcker J, Lennette ET, Fleisher G, Henle W. Primary Epstein-Barr virus infections in African infants. II. Clinical and serological observations during seroconversion. *Int J Cancer*. 1978 Sep 15;22(3):244-50. doi: 10.1002/ijc.2910220305. PMID: 212370.

Brodsky AL, Heath CW Jr. Infectious mononucleosis: epidemiologic patterns at United States colleges and universities. *Am J Epidemiol*. 1972 Aug;96(2):87-93. doi: 10.1093/oxfordjournals.aje.a121444. PMID: 4340252.

Sawyer RN, Evans AS, Niederman JC, McCollum RW. Prospective studies of a group of Yale University freshmen. I. Occurrence of infectious mononucleosis. *J Infect Dis.* 1971 Mar;123(3):263-70. doi: 10.1093/infdis/123.3.263. PMID: 4329526.

Zhou Y, Zhu G, Charlesworth JC, Simpson S Jr, Rubicz R, Göring HH, Patsopoulos NA, Lavery C, Wu F, Henders A, Ellis JJ, van der Mei I, Montgomery GW, Blangero J, Curran JE, Johnson MP, Martin NG, Nyholt DR, Taylor BV; ANZgene consortium. Genetic loci for Epstein-Barr virus nuclear antigen-1 are associated with risk of multiple sclerosis. *Mult Scler.* 2016 Nov;22(13):1655-1664. doi: 10.1177/1352458515626598. Epub 2016 Jan 27. PMID: 26819262.

Kachuri L, Francis SS, Morrison ML, Wendt GA, Bossé Y, Cavazos TB, Rashkin SR, Ziv E, Witte JS. The landscape of host genetic factors involved in immune response to common viral infections. *Genome Med.* 2020 Oct 27;12(1):93. doi: 10.1186/s13073-020-00790-x. PMID: 33109261; PMCID: PMC7590248.

Rubicz R, Yolken R, Drigalenko E, Carless MA, Dyer TD, Bauman L, Melton PE, Kent JW Jr, Harley JB, Curran JE, Johnson MP, Cole SA, Almasy L, Moses EK, Dhurandhar NV, Kraig E, Blangero J, Leach CT, Göring HH. A genome-wide integrative genomic study localizes genetic factors influencing antibodies against Epstein-Barr virus nuclear antigen 1 (EBNA-1). *PLoS Genet.* 2013;9(1):e1003147. doi: 10.1371/journal.pgen.1003147. Epub 2013 Jan 10. PMID: 23326239; PMCID: PMC3542101.

We have revised the manuscript on the 1st paragraph on Page 13, which is shown below.

“We found no association between BL with three GWAS SNPs (all p values > 0.05) that were previously associated with higher anti-EBV EBNA1 IgG antibodies in Europeans (rs2516049) [ref. #38] and Mexican Americans (rs477515 and rs2854275 [ref. #39]) ... ”

4. It's a bit confusing for the methods that the authors employed to gain insights about whether BL associated HLA alleles have effects on *P. falciparum* infection. First, the authors focused on analyses with control samples. The authors claimed no associations of HLA alleles with *P. falciparum* in controls (Fig S9 and S10), but the pathogen infection seems a protective factor for BL (35.3% in cases vs 48.3% in controls). Moreover, it's unclear why the infection was adjusted in the GLMMs reported in Table 2. Any associations remained significant in this joint model would indicate additional contributors to BL risk other than the infection.

RESPONSE: We thank the reviewer for this question. While it is widely accepted that exposure to *P. falciparum* is the strongest geographical cofactor for BL providing a more consistent explanation for geographic variation in BL incidence in Africa (Rainey JJ et al. 2007), it was not clear whether BL risk was related to acute malaria or infection including that without symptoms. Our epidemiological studies have indicated that BL risk is associated with asymptomatic infection (Redmond LS et al., 2020) and that BL cases seem to have robust control of parasitemia that is present before and persists after BL onset, based on reports of lower frequency of fever due to malaria in the past 6 months, lower frequency of detected infection either by microscopy (which measures current infection) or antigenemia (which measures recent infection in the past 4 weeks) (Peprah S et al., 2020). Moreover, we recently showed that BL risk increases by 39% per hundred additional infections and that BL risk is a function of cumulative rather than recent cross-sectional exposure (Broen K et al., 2023), which is normally well controlled. Our conceptual approach to analysis therefore considered *P. falciparum* malaria as the biggest factor that must be assessed carefully. Our analysis focused on HLA alleles previously linked to malaria and used an agnostic approach to search for any other alleles that may be relevant to *P. falciparum* infection in the controls. Although these analyses were uninformative, we still considered *P. falciparum* as a confounder as well and included it in the models to assess such that any associations

remained significant in this joint model would indicate additional contributors to BL risk other than the infection.

Reference list:

Rainey JJ, Mwanda WO, Wairiumu P, Moormann AM, Wilson ML, Rochford R. Spatial distribution of Burkitt's lymphoma in Kenya and association with malaria risk. *Trop Med Int Health*. 2007 Aug;12(8):936-43. doi: 10.1111/j.1365-3156.2007.01875.x. PMID: 17697088.

Redmond LS, Ogwang MD, Kerchan P, Reynolds SJ, Tenge CN, Were PA, Kuremu RT, Masalu N, Kawira E, Otim I, Legason ID, Dhudha H, Ayers LW, Bhatia K, Goedert JJ, Mbulaiteye SM. Endemic Burkitt lymphoma: a complication of asymptomatic malaria in sub-Saharan Africa based on published literature and primary data from Uganda, Tanzania, and Kenya. *Malar J*. 2020 Jul 28;19(1):239. doi: 10.1186/s12936-020-03312-7. PMID: 32718346; PMCID: PMC7385955.

Peprah S, Ogwang MD, Kerchan P, Reynolds SJ, Tenge CN, Were PA, Kuremu RT, Wekesa WN, Sumba PO, Masalu N, Kawira E, Magatti J, Kinyera T, Otim I, Legason ID, Nabalende H, Dhudha H, Ally H, Genga IO, Mumia M, Ayers LW, Pfeiffer RM, Biggar RJ, Bhatia K, Goedert JJ, Mbulaiteye SM. Risk factors for Burkitt lymphoma in East African children and minors: A case-control study in malaria-endemic regions in Uganda, Tanzania and Kenya. *Int J Cancer*. 2020 Feb 15;146(4):953-969. doi: 10.1002/ijc.32390. Epub 2019 May 20. PMID: 31054214; PMCID: PMC6829037.

Broen K, Dickens J, Trangucci R, Ogwang MD, Tenge CN, Masalu N, Reynolds SJ, Kawira E, Kerchan P, Were PA, Kuremu RT, Wekesa WN, Kinyera T, Otim I, Legason ID, Nabalende H, Buller ID, Ayers LW, Bhatia K, Biggar RJ, Goedert JJ, Wilson ML, Mbulaiteye SM, Zelter J. Burkitt lymphoma risk shows geographic and temporal associations with *Plasmodium falciparum* infections in Uganda, Tanzania, and Kenya. *Proc Natl Acad Sci U S A*. 2023 Jan 10;120(2):e2211055120. doi: 10.1073/pnas.2211055120. Epub 2023 Jan 3. PMID: 36595676; PMCID: PMC9926229.

We have revised the manuscript to make our reasoning clearer:

“*P. falciparum* is the strongest known co-factor for the geographic patterns of BL [ref. #25, #31, #32, #33], Thus, it was considered *a-priori*, as a risk factor and a confounder and included in our main models. When interpreted as a confounder, any associations with HLA alleles that remain significant indicates that additional contribution of *P. falciparum* was not responsible for the observed associations with HLA alleles.”

5. The author used “broad groupings of HLA alleles as common (allele frequency [AF] >5%) otherwise as rare to investigate global HLA associations with BL.” According to their QC procedure, which removed variants with MAFs lower than 1%, the “rare” category consisted of variants with MAFs between 1%-5%. With such a range of MAFs, the variants are unlikely considered as rare.

RESPONSE: We appreciate the reviewer’s feedback. We excluded variants that were <1% because of concerns about the sensitivity of Imputation for rare variants. We understand that “rare” has a specific meaning in genetics, so we agree that our usage of “rare” might be confusing. We have revised our manuscript to define the two categories of HLA allele frequency as common if their frequency was >5% and not common when the frequency was <5%. The revision is as shown below:

“Finally, we defined HLA allele groups as common, when the allele frequency [AF] >5%, otherwise as not common when the allele frequency was 1%-5%. Alleles with <1% of the participants were not included in this analysis.”

6. Was the definition of HLA zygosity taking into account the frequencies of HLA alleles?

RESPONSE: The HLA zygosity analysis included all HLA alleles at 2-field resolution regardless of their frequency. This information has been added to the 3rd paragraph on Page 10, as shown below.

“Second, we investigated associations between BL and HLA zygosity at six HLA loci (*HLA - A, B, C and HLA - DRB1, DQB1, and DPB1*). This approach was selected to reduce potential selection bias, as it could arise from excluding individuals carrying rare HLA alleles, particularly given their higher likelihood of being heterozygotes.”

7. Additionally, there are some typos in the manuscript, like “singficaint” in line 101 and “cance” in line 338.

RESPONSE: We are sorry for not carefully proof-reading the manuscript. We have done this in the revised version.

REVIEWERS' COMMENTS:

Reviewer #1 (Remarks to the Author):

The authors made a great effort in responding to the comments, providing careful edits, and including more details on the immunobiology of this disease in the manuscript. Very exciting work.

I found just a couple typos:

- Last sentence of intro needs a space between "rs2040406(G)" and "with".
- On Page 15 - "The 3D structure of HLA-DQA1 chain suggest that Gln 53 is located with a peptide binding groove, therefore this change may be functional" should be "The 3D structure of the HLA-DQA1 chain suggests that Gln 53 is located within the peptide binding groove of HLA-DQ molecules, and may have functional impact on specificity of peptide binding and/or TCR contacts".

Reviewer #2 (Remarks to the Author):

All previous concerns have been fully addressed.